# PPLNs: Parametric Piecewise Linear Networks for Event-Based Temporal Modeling and Beyond

**Chen Song**    **Zhenxiao Liang**    **Bo Sun**    **Qixing Huang**
Department of Computer Science
The University of Texas at Austin
Austin, TX 78712
`{song, liangzx, bosun, huangqx}@cs.utexas.edu`

## Abstract

We present Parametric Piecewise Linear Networks (PPLNs) for temporal vision inference. Motivated by the neuromorphic principles that regulate biological neural behaviors, PPLNs are ideal for processing data captured by event cameras, which are built to simulate neural activities in the human retina. We discuss how to represent the membrane potential of an artificial neuron by a parametric piecewise linear function with learnable coefficients. This design echoes the idea of building deep models from learnable parametric functions recently popularized by Kolmogorov–Arnold Networks (KANs). Experiments demonstrate the state-of-the-art performance of PPLNs in event-based and image-based vision applications, including steering prediction, human pose estimation, and motion deblurring. The source code of our implementation is available at https://github.com/chensong1995/PPLN.

## 1 Introduction

Event cameras are neuromorphic sensors that summarize the evolving world as a stream of *events*. Each event describes the pixel coordinates, time, and polarity of an intensity change. Thanks to the simplicity of this data representation, event cameras enjoy multiple advantages over conventional cameras, including but not limited to fast data rate and high dynamic range (Gallego et al., 2020). Over the past, researchers have developed event-based algorithms to solve various computer vision problems, such as motion deblurring (Pan et al., 2019, 2020; Wang et al., 2020; Song et al., 2022, 2024), human pose estimation (Calabrese et al., 2019), and autonomous driving (Binas et al., 2017; Hu et al., 2020). A recent survey (Zheng et al., 2023) indicates a rapidly increasing community interest in event-based research, with motion deblurring being the most popular task. Extensive experiments demonstrate that by utilizing events as an auxiliary input, algorithms perceive fine motion details that are absent from conventional image captures, leading to substantial performance gain over methods that make inferences from conventional images alone.

The success of event cameras demonstrates the power of imitating biological neuromorphic principles. The event camera comprises a rectangular array of *dynamic vision sensors*, each of which is analogous to a visual receptor neuron on the human retina dedicated to perceiving one specific location. The neuron experiences excitement and produces a spike when the environmental intensity varies significantly, corresponding to an event generated as a response to brightness changes.

This paper presents Parametric Piecewise Linear Networks (PPLNs) to understand the bio-inspired event data with a bio-inspired deep learning model. As opposed to a generic network design, we believe and verify in this paper that it is highly beneficial to build a processing network that caters to the principles of the data source (event cameras). As illustrated by Figure 1 (Middle), the key idea is to explicitly approximate the membrane potential of a neuron as a piecewise linear mapping from time to electric voltage. Figure 1 (Right) presents the sketch of a PPLN node, whose internal mechanism

38th Conference on Neural Information Processing Systems (NeurIPS 2024).

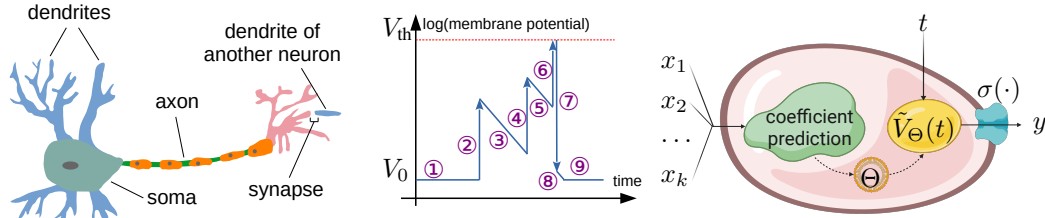

Figure 1: **(Left)**: A biological neuron has three main components: the dendrites (blue), the axon (orange, pink), and the soma (green). The dendrites are responsible for receiving external inputs. The axon transmits signals to the dendrites of other neurons through the synapses. The soma is the body of the cell and connects the dendrites to the axon. **(Middle)**: The membrane potential, defined as the voltage difference between the interior and the exterior of the cell, regulates the neuron's behavior and can be approximately modeled by a piecewise linear function. ① When the neuron is at rest, the potential stays at a constant level $V_0$. ② An external input is received by the dendrites, causing an instantaneous perturbation to the membrane potential. ③ The perturbation is not significant enough to excite the neuron, and the potential leaks over time exponentially (*i.e.*, linearly in the logarithmic space). ④ Another external input happens. ⑤ The input fails to excite the neuron. ⑥ A third input causes the membrane potential to exceed the threshold voltage $V_{th}$. The neuron becomes excited and generates a spike. ⑦ The excitement opens ion channels, and the ion flow causes a reset to the membrane potential. ⑧ After the excitement, the ion channels close again, and the potential continues to decay. ⑨ The neuron returns to the resting state, waiting for new inputs. **(Right)**: A PPLN node. Given inputs $\{x_i\}_{i=1}^k$, we predict the linear coefficients $\Theta$ for the membrane potential function, including the slope, intercept, and endpoints of each line segment. The resulting parametric function $\tilde{V}_\Theta$ is then used to evaluate the neuron output at the timestamp of interest $y(x_1, \ldots, x_k, t) = \sigma(\tilde{V}_\Theta(t))$, where $\sigma(\cdot)$ is the integral normalization defined in Section 3.4.

is explained thoroughly in Section 3. While the event camera imitates how a single layer of visual receptor neurons react to external intensity changes, PPLNs are motivated by observations of how layers of biological neurons communicate. Inspired by the Leaky Integrate-and-Fire model (Abbott, 1999), we propose to use a piecewise linear parameterization to approximate its temporal evolution in the logarithmic space. The key difference between PPLNs and the bio-inspired Spiking Neural Networks (SNNs) (Eshraghian et al., 2021) in existing literature is that PPLNs are an alternative to general GPU-based temporal inference models, whereas SNNs aim at the deployment on hardware neuromorphic chips (Davies et al., 2018). Instead of real-valued membrane potentials, SNNs also propagate binary spikes, leading to reduced energy consumption and training instabilities.

PPLNs are conceptually similar to the emerging Kolmogorov–Arnold Networks (KANs) (Liu et al., 2024). Both KANs and PPLNs leverage learnable parametric functions to build a deep network. Different from KANs, which use input-independent B-splines, PPLNs exploits input-dependent piecewise linear functions. Using piecewise linear functions allows our brain-inspired design to mimic biological neural principles and the event generation model. Predicting function coefficients at inference time allows the network to better handle input heterogeneity.

We modify the network architecture in state-of-the-art event-based vision algorithms by replacing multi-layer perceptions and convolution operators with PPLN nodes and evaluate PPLNs on three applications: steering prediction, 3D human pose estimation, and motion deblurring. With fewer or a similar number of trainable parameters, PPLNs improve baselines by 30.8% in steering prediction, 11.1% in human pose estimation, and 5.6% in motion deblurring. To demonstrate the potential of PPLNs, we experiment on the conventional frame-based version of the same applications without the event input and observe consistent improvements. Additionally, we present a mathematical analysis of the convergence properties to showcase the robustness.

In summary, we make the following contributions:

- We propose Parametric Piecewise Linear Networks (PPLNs), mimicking biological principles by approximating membrane potentials as parametric mappings.
- We show how to predict a set of parametric coefficients from the input of the PPLN node and evaluate the membrane potential at any timestamp of interest.

- We present a mathematical analysis of the convergence properties of PPLNs.
- We apply PPLNs to event-based and frame-based applications and achieve state-of-the-art results.

## 2 Related work

**Biological neurons**. As shown in Figure 1 (Left), if the combined effect of small perturbations over time causes the membrane potential to exceed the threshold potential, the neuron will become excited. During the excitement, the neuron produces an output spike through the synapses (Lacinová, 2005; Maeda et al., 2009). In Figure 1 (Middle), the Leaky Integrate-and-Fire model (Abbott, 1999) suggests that the membrane potential can be approximated by a piecewise linear function as a parametric mapping from time to the logarithm of the voltage difference.

**Spiking neural networks**. Spiking Neural Networks (SNNs) are Recurrent Neural Networks (RNNs) that simulate biological neurons (Eshraghian et al., 2021). Each SNN node carries an internal variable corresponding to the membrane potential. The output of the SNN node is a binary signal that is zero by default and becomes one if excited (Abbott, 1999). The key advantage of SNNs is low power consumption, making them the ideal model to be deployed on hardware neuromorphic chips (Davies et al., 2018). By contrast, PPLNs are designed to be an alternative to general GPU-based temporal inference models, even though SNNs and PPLNs are motivated by similar biological principles.

**Event cameras**. Event cameras are neuromorphic devices that summarize the evolving environment as a stream of events (Lichtsteiner et al., 2008). Each event is analogous to a significant intensity change that exceeds the hardware threshold and excites a biological light-sensing neuron. An event is represented as a 4-tuple $(x, y, t, p)$, where $(x, y)$ are the pixel coordinates, $t$ is the timestamp, and $p \in \{-1, +1\}$ is the polarity of the intensity change. Event cameras have a fast data rate, high dynamic range, low power consumption, and minimal motion blur compared to conventional cameras (Gallego et al., 2020). A recent trend is to utilize hardware that simultaneously captures event and conventional streams, where the event stream is used as an auxiliary input to enhance regular computer vision algorithms. For example, while image-to-image deblurring is a well-studied problem in the research community (Richardson, 1972; Fish et al., 1995; Krishnan & Fergus, 2009; Joshi et al., 2009; Levin et al., 2007; Kim et al., 1998; Shan et al., 2008; Fergus et al., 2006; Xu et al., 2013; Xu & Jia, 2010; Perrone & Favaro, 2014; Babacan et al., 2012; Kupyn et al., 2018, 2019) , several works in event-based vision demonstrate the possibility of converting a blurry image into a sharp video that explains the motion during the exposure interval (Pan et al., 2019, 2020; Wang et al., 2020; Song et al., 2022, 2024).

**Learning activation functions**. The Rectified Linear Unit (ReLU) (Nair & Hinton, 2010) represents a two-piece linear activation function, $f(x) = x$ if $x > 0$, and $f(x) = 0$ if $x \le 0$ and has several variants. Our design is conceptually similar to the Piecewise Linear Unit (PWLU) (Zhou et al., 2021), where the activation is defined as an $n$-piece linear function with learnable slopes and intercepts. Kolmogorov-Arnold Networks (KANs) have also received wide attention, which build a deep model by stacking layers of parametric B-spline activation functions (Liu et al., 2024). In addition to the conceptual similarities, PPLNs are fundamentally different from PWLUs and KANs since PPLNs incorporate temporal modeling. Additionally, PPLNs allow discontinuities at segment endpoints and the learning of endpoint locations, neither of which is supported by PWLUs or KANs. Furthermore, the ReLUs are activation functions to be appended after prediction layers (*e.g.*, linear and convolution layers), whereas the PPLNs are designed to replace the prediction layers in temporal learning models.

## 3 Method

### 3.1 Overview

As shown in Figure 2 (a, b), a PPLN node implements the following mapping:

$$f : \mathbb{R}^k \times [0, 1] \to \mathbb{R} \tag{1}$$

where $\mathbf{x} \in \mathbb{R}^k$ is the $k$-dimensional non-temporal component of the input, and $t \in [0, 1]$ is the normalized input scalar timestamp. The mapping $f$ converts the input $(\mathbf{x}, t)$ to a scalar in $\mathbb{R}$.

The first step in the calculation of $f$ is to predict the linear coefficients, $\Theta = \{\mathbf{m}, \mathbf{b}, \mathbf{s}\}$, using the trainable parameters, $W_m$, $W_b$, $W_s$, and $\mathbf{w}_V$. Section 3.2 explains this process in detail. The

predicted coefficients allow us to assemble the piecewise linear membrane potential function, as illustrated by the blue plot in the bottom-left corner of Figure 2 (a, b). To better handle the numerical instability, Section 3.3 discusses the procedure to smooth the boundaries of the predicted linear pieces. The smoothed function is then normalized by another predicted value $\overline{V}$, as explained in Section 3.4. Finally, the output of the PPLN node is given as $f(\mathbf{x}, t) = \sigma(\tilde{V}_\Theta(t))$ (*i.e.*, the smoothed, and normalized potential function evaluated at the timestamp of interest). While it is straightforward to construct a fully-connected network by stacking PPLN nodes, Section 3.5 discusses how to support convolution operations.

## 3.2 Coefficient prediction

Let $n$ be a hyper-parameter denoting the number of line segments in the piecewise linear modeling. The parametric coefficients $\Theta = \{\mathbf{m}, \mathbf{b}, \mathbf{s}\}$ are given by:

$$\mathbf{m} = \tanh(W_m \cdot \mathbf{x}) \tag{2}$$
$$\mathbf{b} = W_b \cdot \mathbf{x} \tag{3}$$
$$\mathbf{s} = \mathrm{softmax}(W_s \cdot \mathbf{x}) \tag{4}$$

where $W_m$, $W_b$, and $W_s$ are $n \times k$ dimensional matrices containing trainable weights. $\mathbf{m} = (m_1, \ldots, m_n)^T$ and $\mathbf{b} = (b_1, \ldots, b_n)^T$ are the slopes and intercepts of $n$ different line segments, respectively. $\mathbf{s} = (s_1, \ldots, s_n)^T$ defines the temporal interval size for each line segment. Let $t_0 = 0$ and $t_i = t_{i-1} + s_i$. With the softmax function, the temporal space $[0, 1]$ is divided into $n$ non-overlapping intervals by $\mathbf{s}$. We predict the aforementioned coefficients and approximate the membrane potential as:

$$\tilde{V}_\Theta(t) := \begin{cases} m_1 t + b_1 & t_0 \le t < t_1 \\ m_2 t + b_2 & t_1 \le t < t_2 \\ \ldots \\ m_n t + b_n & t_{n-1} \le t \le t_n \end{cases} \tag{5}$$

Here, the hyperbolic tangent function restricts the slope, $\mathbf{m}$, preventing the exploding gradient problem commonly observed in temporal models (Pascanu et al., 2013; Bengio et al., 1994).

## 3.3 Smoothing

While we can build a network by stacking layers of the vanilla PPLN nodes described above, training presents a challenge for numerical stability. Equation (5) suggests that $\frac{\partial \tilde{V}_\Theta(t)}{\partial \mathbf{t}}$ is an all-zero vector. In other words, gradient-based optimizers (Bottou et al., 1991; Kingma & Ba, 2014) cannot update the values of $t_i$'s (the location of segment endpoints).

To address this issue, we propose to smooth the segment boundaries. The key idea is to blend the linear pieces across adjacent intervals. Let $(x, \pi_i(x))$ be the point on the $i^{\text{th}}$ predicted segment, that is, $\pi_i(x) := m_i x + b_i$.

Assuming $t_{i-1} \le t < t_i$ (*i.e.*, $t$ belongs to the $i^{\text{th}}$ predicted segment), the smoothed potential is:

$$\tilde{V}_\Theta^T(t) := w_\leftarrow^{(i)} \pi_{i-1}(t) + (1 - w_\leftarrow^{(i)} - w_\rightarrow^{(i)}) \pi_i(t) + w_\rightarrow^{(i)} \pi_{i+1}(t) \tag{6}$$

where the weights $w_\leftarrow^{(i)}$ and $w_\rightarrow^{(i)}$ are defined through the temperature hyper-parameter $T$:

$$w_l^{(i)} := \begin{cases} 0 & i = 1 \\ \left(1 + \exp\left(T(t - t_{i-1})\right)\right)^{-1} & i = 2, \ldots, n \end{cases}$$

$$w_r^{(i)} := \begin{cases} \left(1 + \exp\left(T(t_i - t)\right)\right)^{-1} & i = 1, \ldots, n - 1 \\ 0 & i = n \end{cases}.$$

Importantly,

$$\lim_{T \to +\infty} \tilde{V}_\Theta^T(t) = \tilde{V}_\Theta(t).$$

The difference between the smoothed potential, $\tilde{V}_\Theta^T(t)$, and the unsmoothed potential, $\tilde{V}_\Theta(t)$, is that the smoothed gradients $\frac{\partial \tilde{V}_\Theta^T(t)}{\partial \mathbf{t}}$ do not vanish. We present a theorem with proof in the appendix

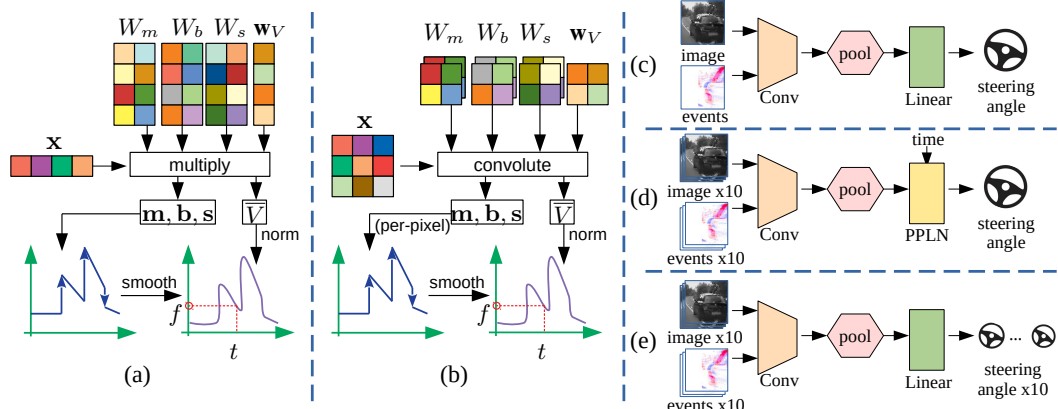

Figure 2: **(a)** A linear PPLN node, which maps the input $(\mathbf{x}, t)$ to output $f$. The trainable parameters are $W_m$, $W_b$, $W_b$, and $\mathbf{w}_V$. **(b)** A similarly structured 2D convolutional PPLN node. **(c)** The baseline architecture for steering angle prediction (Hu). **(d)** Our model. **(e)** The modified baseline (HuMod).

showing the local convergence properties of the piecewise linear model after smoothing. Assuming the underlying function to fit is indeed an $n$-piece piecewise linear function, the theorem states that the coefficients can be accurately learned from a set of noisy samples, provided that the noises are reasonably small, the coefficients are adequately initialized, and the temperature $T$ is sufficiently large. For ease of discussion, the theorem uses segment endpoints $\mathbf{t} = \{t_i\}$ instead of interval lengths $\mathbf{s}$ for the parameterization. Their relation is given in Section 3.2.

**Theorem 3.1.** *(Informal) Consider an underlying n-segment piecewise linear function parameterized by $\Theta^\star = \{\mathbf{m}^\star, \mathbf{b}^\star, \mathbf{t}^\star\}$ as defined in (5). Let $(\tau_j, v_j)$, $j = 1, \ldots, m$ be $m$ point samples, where $v_j = \tilde{V}_{\Theta^\star}(\tau_j) + \psi_j$ in which $\psi_j$ is a small random noise.*

*The L2 loss for the smoothed curve is defined by:*

$$\mathcal{L}^T(\Theta) := \sum_{j=1}^{m} \left(\tilde{V}_\Theta^T(\tau_j) - v_j\right)^2. \tag{7}$$

*Denote by $\Theta_T^\star$ the weights at which the minimum of (7) is attained at temperature $T$. Then we show that starting from some initial $\Theta_0$ close to $\Theta^\star$, by applying vanilla gradient descent with appropriate temperature increase strategy and a learning rate $\eta = O(\frac{1}{T})$, $\Theta$ is guaranteed to converge to $\Theta_\infty^\star$ at a linear convergence rate.*

*Specifically, the error of recovered segments is bounded by:*

$$\sup_{\tau_{\min} \leq \tau \leq \tau_{\max}} |\tilde{V}_{\Theta_\infty^\star}(\tau) - \tilde{V}_{\Theta^\star}(\tau)| < O(\max|\psi_j|), \tag{8}$$

*where $\tau_{\min}$, $\tau_{\max}$ are the smallest and largest values among $\tau_j$, for $j = 1, \ldots, m$, respectively.*

In addition to the theorem above, the supplementary material uses ablation studies to discuss the practical implication of incorporating the smoothing operation.

### 3.4 Integral normalization

While the smoothing operator introduced above enriches temporal gradients, $\frac{\partial \tilde{V}_\Theta(t)}{\partial \mathbf{t}}$, the integral normalization operator addresses the issue that $\frac{\partial \tilde{V}_\Theta(t)}{\partial \mathbf{m}}$ and $\frac{\partial \tilde{V}_\Theta(t)}{\partial \mathbf{b}}$ are both very sparse vectors with only one non-zero entry out of all $n$ elements. From Equation (5), we have:

$$\int_0^1 \tilde{V}_\Theta(t)dt = \frac{1}{2}\sum_{i=1}^{n} m_i(t_i^2 - t_{i-1}^2) + \sum_{i=1}^{n} b_i(t_i - t_{i-1}) \tag{9}$$

Let $\overline{V}$ be a parameter that controls the mean of $\tilde{V}_\Theta(t)$ when $0 \leq t \leq 1$. The integral normalization operator $\sigma(\cdot)$ is defined as:

$$\sigma(\tilde{V}_\Theta(t)) = \tilde{V}_\Theta(t) - \int_0^1 \tilde{V}_\Theta(t)dt + \overline{V} \approx V(t) \tag{10}$$

After the normalization, $\frac{\partial \sigma(\tilde{V}_\Theta(t))}{\partial \mathbf{m}}$ and $\frac{\partial \sigma(\tilde{V}_\Theta(t))}{\partial \mathbf{b}}$ are both dense vectors containing rich gradient information in every element, encouraging a smooth and swift convergence. A side effect of the normalization is that the temporal derivative, $\frac{\partial \sigma(\tilde{V}_\Theta(t))}{\partial \mathbf{t}}$, also becomes non-zero, allowing segment endpoints to be learned even without smoothing.

Notably, the ground-truth parameter $\overline{V}$ is observable in certain applications. In motion deblurring, the task is to generate a sharp video from a blurry image. Mathematically, the mean of all the frames in the output must be equal to the input, restricting the temporal average $\int_0^1 \sigma(\tilde{V}_{\Theta_{xy}}(t))$ to be equal to the input pixel values. When it cannot be easily observed, $\overline{V}$ is regressed from the input $\mathbf{x}$:

$$\overline{V} = \langle \mathbf{w}_V, \mathbf{x} \rangle \tag{11}$$

where $\mathbf{w}_V$ is a $k$-dimensional vector of trainable weights, and $\langle \cdot, \cdot \rangle$ stands for the inner product.

Section 4.4 uses ablation studies to demonstrate the effectiveness of integral normalization. In the appendix, we additionally use a two-piece toy example to analyze the normalization.

### 3.5 Supporting the convolution operation

The above modeling naturally extends to convolutions, which are the fundamental building blocks of contemporary deep learning. In a convolution layer, each input pixel only affects the output in a small spatial neighborhood rather than across the entire grid. To support convolution, the trainable parameters, $W_m$, $W_b$, $W_s$, and $\mathbf{w}_V$, become sparse matrices and vectors with non-zero entries only in locations within the spatial perceptive field. In practice, we predict the coefficients as:

$$\mathbf{m} = \tanh(\text{conv}(W_m, \mathbf{x})) \tag{12}$$
$$\mathbf{b} = \text{conv}(W_b, \mathbf{x}) \tag{13}$$
$$\mathbf{s} = \text{softmax}(\text{conv}(W_s, \mathbf{x})) \tag{14}$$
$$\overline{V} = \text{conv}(\mathbf{w}_V, \mathbf{x}) \tag{15}$$

where, after reshaping, $W_m$, $W_b$, $W_s$, and $\mathbf{w}_V$ are kernels in the convolution operation. The channel-wise softmax operator ensures the temporal interval sizes of each pixel add up to one, which is a requirement posed by the valid range of input timestamps ($t \in [0, 1]$). We refer interested readers to our code release for how the above design is implemented under PyTorch.

## 4 Evaluation

Section 4.1 starts by showing how PPLNs outperform various methods in motion deblurring, the most popular event-based application as indicated by a recent survey (Zheng et al., 2023). In Sections 4.2 and 4.3, we proceed with two other tasks where the goals are to predict the vehicle's steering angle from the dashcam footage and to estimate the 3D human pose from binocular 2D event camera captures. These are two "mainstream" applications in event-based vision, ranking immediately after deblurring, as reported by the survey. Finally, Section 4.4 and the appendix use ablation studies to demonstrate the importance of the integral normalization operator, the effect of changing the number of line segments in the parameterization $n$, as well as the practical implication of the smoothing operator.

We emphasize event-based applications because event cameras and PPLNs are both designed to mimic biological neural principles. Due to the limited availability of high-quality data, event-based vision is an emerging field where modeling plays a more important role than data and the effects of PPLNs can be best demonstrated. However, we also present an evaluation on conventional frame-based tasks to demonstrate the generalizability. Meanwhile, we do not include Spiking Neural Network (SNN) baselines. While SNNs focus on the deployment onto neuromorphic chips, PPLNs are designed to be an alternative to general GPU-based temporal inference models.

### 4.1 Task I: motion deblurring

**Task description**. Event-enhanced motion deblurring is a popular research domain. Variants of the task include image-to-image and image-to-video deblurring. Our experiments focus on the highly challenging image-to-video problem, establishing a thorough competition against various state-of-the-art approaches. Given a blurry image and its associated events during the exposure interval, our

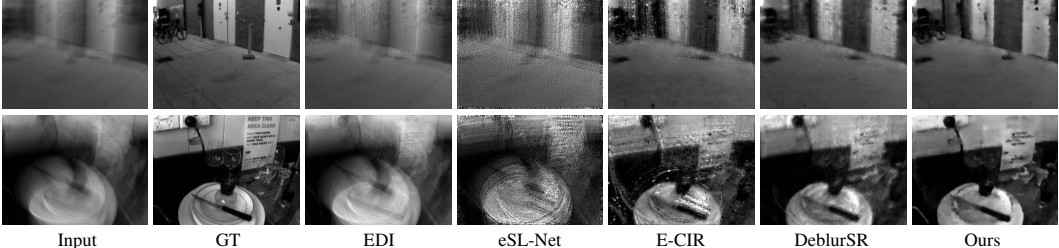

| Input | GT | EDI | eSL-Net | E-CIR | DeblurSR | Ours |

Figure 3: Motion deblurring visualizations. More are available in the supplementary material.

goal is to reverse the exposure process and reconstruct a sharp video describing the relative motion between the camera and the environment. The following experiments utilize the High Quality Frames (HQF) (Stoffregen et al., 2020) dataset and the preprocessing procedure documented by Song et al. We construct a PPLN based on the U-Net architecture (Ronneberger et al., 2015) by replacing all 2D convolution layers with PPLN nodes (Figure 2 (b)). The input to the PPLN is the concatenation of the blurry image and the event histograms, and the histograms are constructed following Zhu et al.. The output of the PPLN is sharp frames at 14 uniformly spaced timestamps. We utilize ADAM (Kingma & Ba, 2014) to train the network for 50 epochs using the L1 loss. We set the learning rate to $10^{-3}$ and reduce the rate by half after 20 and 40 epochs, respectively. The number of line segments is $n = 3$.

**Evaluation metric**. We use the Mean Squared Error, the Peak Signal-to-Noise Ratio, and the Structural Similarity Index Measure.

**Results and discussions**. As shown in Table 1 (Left), PPLN has a very strong performance in motion deblurring. PPLN improves DeblurSR (Song et al., 2024), the state-of-the-art event-based motion deblurring model at the time of paper submission, by 5.6% in MSE, 0.372 dB in PSNR, and 4.9% in SSIM. Importantly, we underscore that the PPLN is a generic network architecture and does not require task-specific modeling, whereas the baseline approaches utilize techniques that cater to the deblurring problem, such as dictionary learning (eSL-Net) (Wang et al., 2020), per-pixel polynomial approximation (E-CIR) (Song et al., 2022), and implicit neural representation (DeblurSR) (Song et al., 2024). The success of the PPLN reveals the strength of mimicking biological neural behaviors.

In Table 1 (Left), we also use regular convolution layers to construct the U-Net. We increase the number of convolutional channels in the regular U-Net to approximately match the number of trainable parameters in the PPLN (173M versus 192M). The last two rows suggest that the PPLN improves the U-Net by 54.5% in MSE, 6.45 dB in PSNR, and 43.3% in SSIM. Qualitatively, as shown in Figure 3, our method generates sharper and more realistic frames than the baseline approaches. The proposed PPLN offers vivid details around salient features. In the first row, our method reconstructs the dark patterns on the white background with sharp edges. In the second row, our method gives the best contrast between the white board and the black letters.

### 4.2 Task II: steering angle prediction

**Task description**. The DAVIS Driving Dataset released in 2020 (DDD20) (Hu et al., 2020) contains 51 hours of dashcam recordings with both neuromorphic events and conventional frames. Following Hu et al., we select 15 recordings during the day and 15 at night across the western United States, and train a deep network to regress steering angles from the dual-modal input.

Table 1: **(Left)**: Motion deblurring quality. **(Right)**: Steering prediction errors.

|  | MSE ↓ | PSNR ↑ | SSIM ↑ |
|---|---|---|---|
| image-to-video baselines | | | |
| EDI | 0.336 | 17.822 | 0.515 |
| eSL-Net | 0.452 | 14.938 | 0.282 |
| eSL-Net++ | 0.385 | 16.870 | 0.363 |
| E-CIR | 0.207 | 21.713 | 0.609 |
| DeblurSR | 0.161 | 23.912 | 0.694 |
| U-Net | 0.334 | 17.834 | 0.508 |
| Ours | **0.152** | **24.284** | **0.728** |

|  | RMSE ↓ | | |
|---|---|---|---|
|  | night | day | all |
| Hu | 3.05 ± 0.104 | 5.71 ± 0.334 | 4.55 ± 0.236 |
| HuMod | 2.64 ± 0.036 | 3.94 ± 0.069 | 3.34 ± 0.048 |
| Ours | **2.53 ± 0.040** | **3.68 ± 0.150** | **3.15 ± 0.081** |
|  | EVA ↑ | | |
|  | night | day | all |
| Hu | 0.940 ± 0.004 | 0.713 ± 0.033 | 0.845 ± 0.016 |
| HuMod | 0.954 ± 0.001 | 0.864 ± 0.005 | 0.917 ± 0.003 |
| Ours | **0.958 ± 0.001** | **0.881 ± 0.010** | **0.926 ± 0.004** |

Table 2: **(Left)**: Human pose estimation errors. **(Right)**: Frame-based steering prediction errors.

| | 2D MPJPE ↓ | | 3D MPJPE ↓ |
| --- | --- | --- | --- |
| | Cam 2 | Cam 3 | |
| Calabrese | 7.49 | 7.29 | 82.17 |
| CalabreseMod | 11.90 | 11.78 | 130.24 |
| Ours | **6.76** | **6.51** | **73.05** |

| | all RMSE ↓ | all EVA ↑ |
| --- | --- | --- |
| Hu | $5.53 \pm 0.110$ | $0.771 \pm 0.009$ |
| HuMod | $3.55 \pm 0.154$ | $0.907 \pm 0.007$ |
| Ours | $\mathbf{3.16 \pm 0.130}$ | $\mathbf{0.927 \pm 0.005}$ |

As shown in Figure 2 (c), we build a PPLN upon the existing baseline, consisting of a convolutional head, a pooling layer, and a linear component. The convolutional head reduces the spatial dimension of the data and increases the number of channels. The pooling layer eliminates both spatial dimensions by averaging all the pixels on the reduced spatial grid. Finally, the linear component of the network maps the 64-dimensional feature to the scalar steering angle output. The network contains a total number of 463,425 (463K) parameters. The input frame and events contain 50 ms of historical data.

Our model (Figure 2 (d)) lowers the prediction frequency to every 500 ms. The network takes ten conventional frames and ten times as many events as input and predicts the steering angle at ten uniformly distributed timestamps. We replace the linear component with PPLN nodes and slightly increase the number of layers. To match the number of parameters in the original architecture, we shrink the convolutional head. This results in a network with 455,338 (455K) parameters. We utilize ADAM (Kingma & Ba, 2014) to train the network for 200 epochs using the L2 loss. The learning rate $10^{-3}$ with a weight decay of $10^{-4}$. The number of line segments in the parameterization is $n = 3$.

**Evaluation metric**. We use the Root Mean Square Error and the Explained VAriance. We train with five random seeds and report the mean and standard deviation.

**Results and discussion**. As shown in Table 1 (Right), our approach outperforms the baseline model in RMSE, with a 17.0% improvement at night, 35.6% improvement during the day, and 30.8% improvement overall. Similar enhancement is observed in EVA, with 1.9% improvement at night, 23.6% improvement during the day, and 9.6% improvement overall.

The input to our model contains ten times as much information as the baseline approach (Hu et al., 2020). To investigate whether the performance gain is simply a result of enriched input information, we present an additional comparison where the baseline is modified to have the same input and output dimensions as our model (Figure 2 (e)). From the second and the third rows in Table 1 (Right), we observe that PPLN improves the modified baseline by 5.7% in RMSE and 1.0% in EVA.

### 4.3 Task III: human pose estimation

**Task description**. The dynamic Vision Sensor Human Pose (DHP19) dataset (Calabrese et al., 2019) is collected by inviting human subjects into a cubic space and using event cameras in the four ceiling corners to record various body movements, such as walking, jumping, and walking. DHP19 contains recordings of 17 human subjects performing 33 different body movements. In addition to the events, DHP19 includes the 3D coordinates of 13 body joints. The goal is to predict 3D joint coordinates.

Calabrese *et al.* use two frontal cameras ("Cam 2" and "Cam 3") in their experiments. The overall pipeline has two stages. First, they utilize a deep network to predict the 2D joint coordinates from the events in each view. After that, they project the 2D predictions into 3D using the calibration matrices.

The deep network used by Calabrese *et al.* is a fully convolutional network (Long et al., 2015) that contains 17 layers and 218,592 (219K) trainable parameters. The input events are represented as histograms (Zhu et al., 2019), and the output 2D joint coordinates are represented as heatmaps. After collecting 25,000 events from all four views, the algorithm assembles event histograms from two frontal cameras and discards the events from the other two cameras. The model updates the 3D joint coordinates if the 2D prediction confidence exceeds a threshold $\tau = 0.3$ in both views.

In our experiment, the model makes an inference every 250,000 events. The network takes ten times as many events as input and predicts 2D joint coordinates at ten different timestamps. We modify the prediction network by introducing PPLN layers. The modified network contains 215,648 (216K) trainable parameters. We utilize RMSProp (Kingma & Ba, 2014) to train the network for 20 epochs using the L2 loss. The learning rate is $10^{-3}$ in the first 10 epochs, $10^{-4}$ from epochs 10 to 15, and $10^{-5}$ from epochs 15 to 20. The number of line segments in the parameterization is $n = 3$.

**Evaluation metric**. We use Mean Per Joint Position Error and report in 2D pixels and 3D millimeters.

Table 3: Ablation studies justifying normalization. The appendix discusses the number of segments.

| Norm? | Motion Deblurring | | | Steering Prediction | | Human Pose Estimation | | |
|---|---|---|---|---|---|---|---|---|
| | MSE ↓ | PSNR ↑ | SSIM ↑ | RMSE ↓ | EVA ↑ | 2D-2 ↓ | 2D-3 ↓ | 3D ↓ |
| ✗ | 0.264 | 19.721 | 0.554 | 3.82 ± 0.339 | 0.897 ± 0.018 | 6.81 | 6.74 | 74.88 |
| ✓ | **0.152** | **24.284** | **0.728** | **3.15 ± 0.081** | **0.926 ± 0.004** | **6.76** | **6.51** | **73.05** |

**Results and discussion**. As shown in Table 2 (Left), PPLN estimations have a 2D MPJPE of 6.76 pixels in Cam 2, a 2D MPJPE of 6.51 pixels in Cam 3, and a 3D MPJPE of 73.05 mm. Compared to the original network used by Calabrese *et al.*, we achieve 9.7% improvement in Cam 2, 10.7% improvement in Cam 3, and 11.1% improvement in 3D. Similar to Section 4.2, simply enlarging the temporal horizon confuses the network and leads to performance degradation. Our method enhances the modified baseline by 43.2% in Cam 2, 44.7% in Cam 3, and 44.0% in 3D.

## 4.4 Ablation studies

In Section 3.4, we introduce the integral normalization operator, which theoretically stabilizes training by enriching the gradient information. We now use ablation studies to examine the effectiveness of integral normalization in practice. As shown in Table 3, the introduction of this operator improves motion deblurring quality by 42.4%, 4.563 dB, and 31.4% in MSE, PSNR, and SSIM respectively. For steering prediction, integral normalization improves the accuracy by 17.5% in MSE and 3.2% in EVA. The p-values from one-tailed t-tests are 0.007 and 0.014 for MSE and EVA, giving us reasonable confidence that integral normalization has enhanced accuracy. For human pose estimation, integral normalization improves the 2D MPJPE by approximately 0.2 pixels and the 3D MPJPE by 1.8 mm. The improvement suggests that integral normalization can effectively regularize the training in scenarios such as deblurring where the normalization target has a clear semantic meaning. We refer readers to the appendix for the impact of line segment number $n$ and smoothing on performance.

## 4.5 Conventional frame-based vision

To demonstrate the generalizability of PPLNs, we remove the event input from the steering angle prediction model. Table 2 (Right) shows that PSNNs can still outperform the baselines in the conventional frame-based only setting. Importantly, we observe a significant performance drop by taking out the event input from both baseline approaches. However, the frame-based PPLN has a surprisingly similar prediction quality to the dual-modal PPLN. This result demonstrates that by simulating the biological behaviors, our model can effectively overcome the imperfections in the input data. Note that among the three tasks discussed above, only steering prediction allows inference from conventional frames alone. Human pose estimation takes events as the single-modal input, and image-to-video deblurring requires events to address the motion ambiguity.

# 5 Conclusion and Future Work

This paper presents Parametric Piecewise Linear Networks (PPLNs), a novel temporal learning architecture inspired by biological neural principles. The key idea is to represent the membrane potential as a parametric piecewise linear function with predictable coefficients. Experiments on various event-based vision applications, including steering prediction, human pose estimation, and motion deblurring, demonstrate that PPLNs outperform state-of-the-art models. In the future, we plan to use a recurrent prediction model to support a dynamic number of line segments. Another direction is to adopt more accurate modeling for the membrane potential function, including mechanisms such as the refractory period after each spike.

# 6 Acknowledgement

Q.H. would like to acknowledge NSF IIS 2047677 and NSF IIS 2413161.

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

# A  Appendix / supplemental material

## A.1  Detailed results for task III

We report the 3D MPJPE in millimeters for all 33 different types of body movements evaluated on 5 testing subjects. As shown in Table 4 and Table 5, simply increasing the input horizon leads to an overall performance degradation. One possible explanation is that the network becomes confused about the temporal relationship among the input events. The modified baseline is not a time series model and can only infer the temporal relation from the input stacking order (the first temporal event slice is in the first channel, the second temporal event slice is in the second channel, etc.). On the other hand, PPLN keeps track of the timestamp of interest in every model layer. This allows PPLN to outperform the baseline in almost all cases by a significant margin. We point out that in Table 4 and Table 5, the asterisk for testing subject 2 in movement 29 indicates missing data in the DHP19 dataset (Calabrese et al., 2019).

Table 4: Baseline 3D MPJPE (mm) for all 33 movements (Mov) across 5 testing subjects (S1-S5).

| Mov | Calabrese | | | | | CalabreseMod | | | | |
|-----|-------|--------|--------|--------|--------|--------|--------|--------|--------|--------|
|     | S1 | S2 | S3 | S4 | S5 | S1 | S2 | S3 | S4 | S5 |
| 1 | 129.98 | 106.31 | 76.50 | 208.78 | 150.00 | 93.10 | 131.08 | 80.60 | 126.23 | 141.42 |
| 2 | 55.99 | 102.24 | 114.81 | 209.18 | 185.36 | 112.76 | 143.94 | 182.98 | 184.06 | 150.85 |
| 3 | 62.37 | 101.59 | **87.35** | 73.37 | 91.80 | 119.69 | 181.39 | 127.62 | 117.21 | 104.18 |
| 4 | 80.61 | **97.78** | 77.96 | 79.43 | **75.53** | 134.74 | 196.88 | 114.39 | 144.19 | 162.52 |
| 5 | 218.86 | 110.42 | 131.30 | 184.56 | 107.39 | 93.58 | **85.00** | 108.69 | **112.07** | 115.57 |
| 6 | 129.51 | 127.40 | 110.65 | 284.49 | 180.67 | 134.96 | 121.44 | 199.31 | 149.55 | 123.78 |
| 7 | 65.64 | 77.11 | 67.41 | 88.31 | 78.75 | 96.42 | 100.02 | 83.96 | 106.65 | 86.91 |
| 8 | 55.38 | 80.06 | 68.16 | 72.23 | 73.87 | 73.71 | 100.67 | 88.75 | 95.25 | 95.17 |
| 9 | 27.02 | **58.79** | 44.59 | 67.95 | **57.86** | 50.73 | 100.72 | 75.09 | 97.14 | 89.23 |
| 10 | 44.20 | **41.18** | **44.48** | 74.10 | 133.35 | 143.43 | 86.08 | 155.72 | 134.12 | 221.21 |
| 11 | 69.24 | 53.57 | 54.84 | 80.80 | **112.31** | 176.06 | 117.72 | 136.26 | 142.60 | 212.07 |
| 12 | **27.05** | 47.68 | **43.23** | 75.01 | 55.49 | 63.81 | 74.67 | 98.47 | 118.29 | 84.77 |
| 13 | 43.24 | **47.00** | 51.16 | 70.75 | 55.70 | 83.29 | 91.36 | 98.98 | 114.56 | 74.45 |
| 14 | **29.65** | 47.56 | 44.99 | 70.29 | 58.73 | 69.81 | 89.07 | 90.21 | 113.56 | 76.72 |
| 15 | 78.41 | 152.67 | 145.61 | 176.31 | 115.96 | 73.83 | 236.78 | 119.71 | 138.51 | 110.47 |
| 16 | 163.25 | 171.15 | 121.49 | 171.82 | 115.15 | 89.90 | 333.42 | 126.44 | 180.29 | 131.83 |
| 17 | 76.99 | 157.57 | 103.21 | 122.52 | 104.77 | 125.37 | 160.45 | 100.94 | 159.16 | 100.92 |
| 18 | 93.29 | 134.34 | 121.31 | 120.69 | 118.20 | 100.01 | 222.67 | 176.49 | 172.93 | 133.98 |
| 19 | 74.57 | **131.33** | 102.23 | 114.75 | 101.88 | 66.02 | 268.41 | 90.25 | 111.69 | 94.62 |
| 20 | 116.45 | **102.90** | 78.70 | 112.17 | 114.54 | 88.80 | 171.14 | 75.72 | 155.22 | 96.05 |
| 21 | 34.04 | **54.99** | **57.06** | **68.36** | 56.26 | 77.92 | 97.53 | 121.81 | 113.11 | 102.91 |
| 22 | 59.59 | 74.76 | 62.84 | 100.41 | 77.84 | 210.92 | 168.00 | 174.50 | 242.17 | 219.59 |
| 23 | 99.88 | 152.25 | 120.68 | 103.85 | 99.92 | 96.98 | 172.55 | 168.81 | 112.61 | 113.61 |
| 24 | 105.15 | 131.17 | 125.13 | 96.49 | 93.74 | 112.49 | 145.38 | 167.80 | 119.84 | 94.63 |
| 25 | 99.21 | 124.79 | 173.60 | 82.97 | 128.60 | 153.60 | 192.03 | 235.51 | 123.17 | 168.88 |
| 26 | 96.56 | 109.12 | 155.25 | 87.75 | 134.26 | 126.87 | 184.11 | 185.12 | 112.14 | 168.00 |
| 27 | 61.77 | 159.26 | 105.29 | 139.66 | 123.18 | 68.55 | 96.35 | 64.91 | 132.84 | 94.20 |
| 28 | 75.63 | 116.32 | 168.53 | 133.53 | 110.60 | 71.64 | 123.16 | 83.89 | 140.78 | 118.35 |
| 29 | 68.34 | * | 101.05 | 220.31 | 159.51 | 109.18 | * | 93.69 | 107.75 | 94.93 |
| 30 | 80.43 | 111.21 | 107.29 | 157.46 | 134.06 | 119.36 | 117.88 | 80.42 | 166.10 | 120.45 |
| 31 | 72.11 | 97.21 | 107.93 | 121.62 | 115.68 | 88.70 | **73.28** | **84.35** | 92.15 | 152.39 |
| 32 | 78.22 | 159.89 | 114.66 | 153.90 | 124.55 | 78.31 | 129.95 | 80.68 | 95.26 | 112.57 |
| 33 | 77.70 | 101.20 | 173.90 | 136.04 | 135.47 | 81.68 | 130.34 | 157.63 | 142.18 | 110.33 |
| Mean | 62.23 | 82.71 | 79.56 | 94.13 | 82.17 | 116.63 | 129.81 | 130.46 | 132.89 | 137.60 |

## A.2  Additional motion deblurring visualizations

Figure 4 presents seven additional examples for the motion deblurring results. We also encourage readers to watch the supplementary animations in the .gif format, which demonstrate the temporal smoothness of our results. Overall, the visual quality of PPLN reconstructions is on par with the state-of-the-art method (DeblurSR (Song et al., 2024)). Quantitatively, PPLN outperforms DeblurSR by a small margin (see paper body). We point out that while all baseline approaches are specifically

Table 5: PSNN's 3D MPJPE (mm) for all 33 movements (Mov) across 5 testing subjects (S1-S5).

| | Ours | | | | |
|---|---|---|---|---|---|
| Mov | S1 | S2 | S3 | S4 | S5 |
| 1 | **45.21** | **53.92** | **50.51** | 97.34 | 100.01 |
| 2 | **38.14** | **59.32** | 72.31 | 114.74 | 104.61 |
| 3 | **58.61** | **98.07** | 89.94 | **70.99** | **85.94** |
| 4 | **59.11** | 107.57 | **68.32** | **72.22** | 81.14 |
| 5 | **51.39** | 88.91 | **101.32** | 120.83 | **80.06** |
| 6 | **76.07** | **88.30** | **76.61** | 114.44 | 109.65 |
| 7 | **48.38** | **65.89** | **62.70** | 80.92 | 72.89 |
| 8 | **47.00** | **69.42** | **63.82** | **68.31** | **68.48** |
| 9 | **26.96** | 59.52 | **43.89** | **61.84** | 58.18 |
| 10 | **44.16** | 44.46 | 48.24 | **71.71** | 130.15 |
| 11 | **66.40** | **51.82** | **53.39** | **78.02** | 113.78 |
| 12 | 27.17 | 47.39 | 44.84 | **74.25** | **54.05** |
| 13 | **42.66** | **47.47** | **48.26** | **67.90** | **52.92** |
| 14 | 32.54 | **45.69** | 42.11 | **65.85** | **56.23** |
| 15 | **63.60** | **141.86** | **88.23** | 120.45 | 101.50 |
| 16 | **60.67** | **162.74** | **83.70** | 121.69 | 108.83 |
| 17 | **62.47** | **152.04** | **77.33** | 111.53 | **88.25** |
| 18 | **82.96** | **113.02** | **78.43** | 114.36 | **93.18** |
| 19 | **58.50** | 135.82 | **78.28** | **95.76** | **82.82** |
| 20 | **58.75** | 117.22 | **65.94** | 100.34 | **76.95** |
| 21 | **32.30** | 62.41 | 59.28 | 68.59 | **55.58** |
| 22 | **52.19** | **72.31** | **62.70** | 99.06 | **73.88** |
| 23 | **80.53** | **117.88** | **111.23** | 92.39 | **83.53** |
| 24 | **85.01** | **119.82** | **124.99** | 89.69 | **69.63** |
| 25 | **94.64** | **106.63** | **168.56** | 78.97 | 113.24 |
| 26 | **96.09** | **90.18** | **133.45** | 80.44 | 104.53 |
| 27 | **42.82** | **87.68** | **54.94** | 84.40 | 77.92 |
| 28 | **51.48** | **78.21** | **56.26** | 88.34 | 79.03 |
| 29 | **51.48** | * | **80.17** | 68.89 | 78.50 |
| 30 | **68.61** | **70.03** | **68.89** | 107.84 | 85.72 |
| 31 | **52.56** | 88.16 | 86.70 | **87.64** | 83.46 |
| 32 | **54.93** | **79.87** | **73.45** | 95.04 | 88.60 |
| 33 | **63.91** | **80.45** | 101.89 | 111.04 | **73.66** |
| Mean | **52.93** | **75.11** | **71.48** | **80.88** | **79.54** |

designed for motion deblurring, the proposed PPLN is a general framework that can be applied in various event-based vision tasks.

## A.3 The neuromorphic mechanism

Some properties of the proposed model are slightly different from the standard practice in existing research in computer vision and machine learning (i.e., Spiking Neural Networks, SNNs (Maass, 1997)), even though both of them are inspired by biological neural principles (Lapicque, 1907).

First, existing SNNs focus on the interconnection between artificial neurons, with different layers communicating through binary signals. On the other hand, PPLN focuses on representing the membrane potentials, defined as the voltage differences between the interior and the exterior of the cell. The membrane potentials change with time according to the piecewise linear representation, whose parameters are predicted from the input.

Second, the signal transmitted between SNN layers carries a one if the neuron is excited and a zero if the neuron is not excited. In contrast, our work models the membrane potential using a real value. When a spike occurs, the neuron becomes excited, and the real-valued membrane potential function experiences a discontinuous gap.

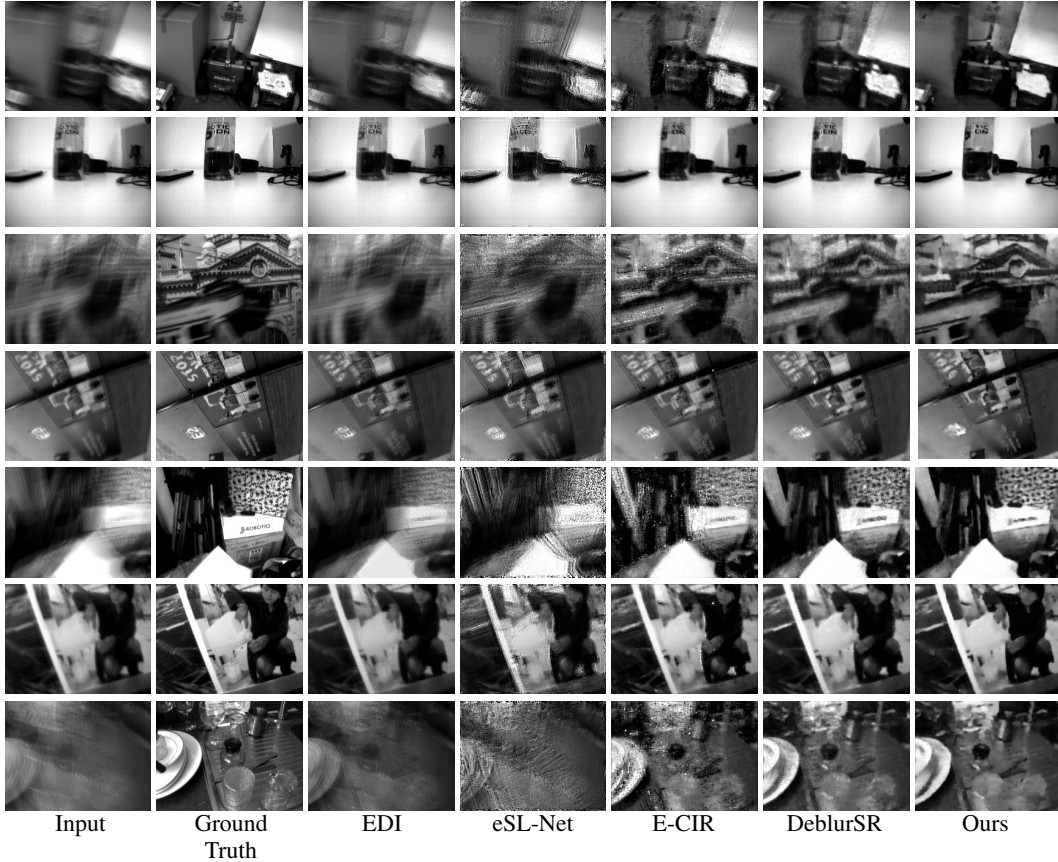

| Input | Ground Truth | EDI | eSL-Net | E-CIR | DeblurSR | Ours |

Figure 4: Motion deblurring visualizations on the HQF dataset.

Third, existing SNNs utilize the linear decay mechanism. In other words, the membrane potential function only decreases in the absence of excitement. This contrasts our parametric mechanism where the slope of each line segment can be either positive or negative. We point out that in biological neurons, the resting potential is not the "absolute" zero. When a biological neuron is at rest, there is naturally a voltage difference between the interior and the exterior of the cell. If the membrane potential exceeds the resting potential, the neuron will decrease the potential at an approximately linear rate. Similarly, if the membrane potential drops below the resting potential, the neuron will gradually increase the potential. The linear potential increment corresponds to a positive slope in the proposed mechanism.

Fourth, we do not explicitly enforce any line segment to be flat. Instead, we expect the network to implicitly learn when a line segment should have a zero slope from the training dataset. This allows more flexibility in the design.

### A.4 Coefficient visualization

In figure 5, we plot a few randomly sampled piecewise linear functions predicted by the network. We observe uneven segment lengths and discontinuities (shown in orange) at boundaries. The piece-wise linear function has varying slopes, suggesting the model does not collapse to somewhere far away from the design. We also observe that with $n = 3$, the network has the capability to represent functions with less than 3 linear segments.

### A.5 Baseline implementation

Despite our best efforts, we fail to exactly reproduce the baseline performance reported by Hu et al. for steering angle prediction and Calabrese et al. for human pose estimation. According to Hu

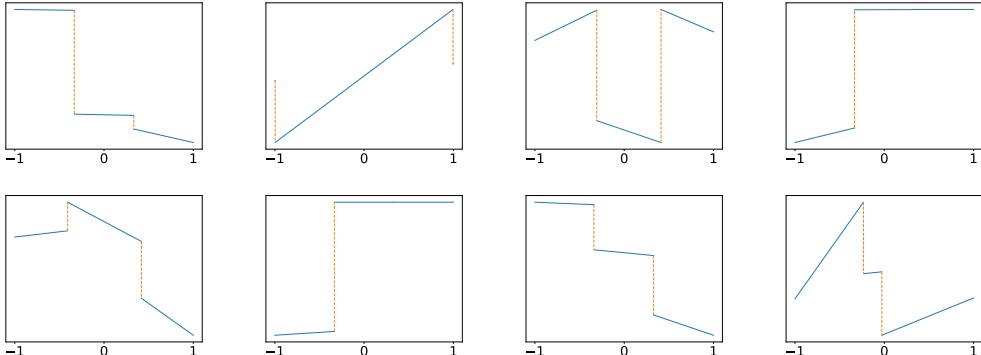

Figure 5: Randomly sampled piecewise linear predictions.

et al., the baseline model has an RMSE of $4.13 \pm 0.24$ and an EVA of $0.881 \pm 0.009$. Our baseline implementation gives an RMSE of $4.55 \pm 0.24$ and an EVA of $0.845 \pm 0.016$. According to Calabrese et al., the baseline 3D MPJPE is 79.63 mm. Our implementation gives 82.17 mm. Similar confusions have been reported by various teams. We refer readers to the issue pages on their GitHub repositories for detailed discussions.

Compared to the relative improvement brought by PPLNs, the differences above are insignificant. We believe the inconsistency in steering angle prediction is the result of ambiguities in the data pruning procedure discussed in Section III-A of the original paper by Hu et al.. We provide step-by-step instructions for the data pruning procedure used when generating our reported figures in the code release. We believe training randomness and differences in optimizer hypermeters cause inconsistency in human pose estimation. Our code release includes the random seed, optimizer hypermeters, and the pre-trained weights for both the baseline model and the PPLN.

## A.6    Data

In steering angle prediction, we use the DDD20 dataset (Hu et al., 2020) released under GNU Lesser General Public License v3.0. In human pose estimation, we use the DHP19 dataset (Calabrese et al., 2019). The DHP19 dataset is released under the Creative Commons Attribution-ShareAlike 4.0 International License, and the dataset utility scripts are released under MIT License. In motion deblurring, we use the HQF dataset (Stoffregen et al., 2020). At the time of paper writing, the HQF dataset is available for public download, but the licensing details are unclear. To the best of our knowledge, the datasets do not contain personally identifiable information or offensive content.

## A.7    Computation resources

We train the model for steering angle prediction using one NVIDIA TITAN V GPU. We train the model for human pose estimation using one NVIDIA Tesla V100 SXM2 GPU. We train the model for motion deblurring using one NVIDIA Tesla V100 SXM2 GPU.

## A.8    Limitations

In this paper, we define the number of line segments in the piecewise linear approximation to the membrane potential function as a hyperparameter $n$. The choice of this hyperparameter affects the representation capacity of the PPLN node, as well as the run-time complexity and convergence rate. At the moment, we are unable to provide a theoretical guideline that allows developers to choose its value based on the input properties and output requirements. Hyperparameter tuning is needed to balance the prediction quality and computational cost.

## A.9 Proof of Theorem 3.1

**Theorem 3.1** Consider an underlying $n$ segment piecewise linear function parameterized by $\Theta^\star = \{\mathbf{m}^\star, \mathbf{b}^\star, \mathbf{t}^\star\}$. Let $(\tau_j, v_j)$, $j = 1, \ldots, m$ be $m$ point samples, where $v_j = \tilde{V}_{\Theta^\star}(\tau_j) + \psi_j$, and $\psi_j$ is a small random noise.

Recall that the ground-truth membrane potential was defined by:

$$\tilde{V}_{\Theta^\star}(t) := \begin{cases} m_1^\star t + b_1^\star & t_0^\star \leq t < t_1^\star \\ m_2^\star t + b_2^\star & t_1^\star \leq t < t_2^\star \\ \ldots \\ m_n^\star t + b_n^\star & t_{n-1}^\star \leq t \leq t_n^\star \end{cases} \tag{16}$$

Without losing generality, we assume the following constraints:

$$t_0^\star = \min_{1 \leq j \leq m} \tau_j = 0$$
$$t_n^\star = \max_{1 \leq j \leq m} \tau_j = 1.$$

The following sampling assumption will be put in order to prevent uncontrollable error due to noise.

---
**Uniform Sampling Assumption:** We assume there exists some constant $c > 0$ such that the $n_i$ samples $(\tau_{i1}, v_{i1}), \cdots, (\tau_{in_i}, v_{in_i})$ on the $i^{\text{th}}$ segment satisfy

$$\frac{\sum_{p<q} |\tau_{ip} - \tau_{iq}|}{\sum_{p<q} (\tau_{ip} - \tau_{iq})^2} \leq c / \max_{p,q} |\tau_{ip} - \tau_{iq}|.$$
---

Then under the objective:

$$\mathcal{L}^T(\Theta) = \sum_{j=1}^m \left( \tilde{V}_{\Theta}^T(\tau_j) - v_j \right)^2 \tag{17}$$

and by applying vanilla gradient descent, $\Theta^\star$ can be recovered up to an error:

$$\max_{0 \leq t \leq 1} \left| \tilde{V}_{\Theta_\infty}^T(t) - \tilde{V}_{\Theta^\star}(t) \right| \leq O\left( \max_j |\psi_j| \right) \tag{18}$$

if the learning rate $\eta = O\left(\frac{h}{n}\right)$, where parameter $h > 0$ measures the distance from the initialization to the ground truth:

$$|t_i(\Theta_0) - t_i^\star| \leq h \leq |t_i^\star - t_{i-1}^\star| \quad \forall i = 1, \ldots, n$$

**Comment:** The uniform sampling assumption is necessary. Imagine a scenario where a disproportionately large number of samples are clustered around $(\tau, v) = (0, 0)$, with only a single sample at $(\tau, v) = (1, 0)$. In such an extreme case, the linear regression model might yield a significantly steep slope due to noise, leading to substantial errors when predicting for $\tau = 1$. This example highlights the potential pitfalls of non-uniform sampling in affecting the reliability of regression outcomes. In addition, it can be shown that a set of uniformly random samples would provide constant $c$ with high probability.

*Proof.* Let $\Theta_r$ be the coefficients in the $r^{\text{th}}$ iteration of the optimization. We first consider the case where all samples $(\tau_j, v_j)$'s are classified into the correct intervals (*i.e.*, $t_{i-1}(\Theta_r) \leq \tau_j < t_i(\Theta_r)$ implies $t_{i-1}^\star \leq \tau_j \leq t_i^\star$). With a sufficiently large temperature $T$, the effect of smoothing becomes minimal, and the estimated potential is:

$$\tilde{V}_{\Theta}^T(t) = (m_i t + b_i)\left(1 + o(\exp(-T))\right)$$

for all $t_{i-1} \leq t < t_i$.

On the other hand, if $t_{i-1}^\star \leq \tau_j < t_i^\star$, the noisy sample is:

$$v_j = m_i^\star \tau_j + b_i^\star + \psi_j.$$

This gives the objective as:

$$\mathcal{L}^T(\Theta) = \sum_{j=1}^{m} \left( \tilde{V}_{\Theta}^T(\tau_j) - v_j \right)^2$$

$$= \sum_{i=1}^{n} \sum_{t_{i-1}^* \leq \tau_j < t_i^\star} \Big( \big( (m_i \tau_j + b_i)(1 + o(\exp(-T))) \big)$$

$$- \big( m_i^\star \tau_j + b_i^\star + \psi_j \big) \Big)^2$$

$$= \sum_{i=1}^{n} \sum_{t_{i-1}^* \leq \tau_j < t_i^\star} \Big( \big( (m_i - m_i^\star)\tau_j + (b_i - b_i^\star) \big)$$

$$- \psi_j + o(\exp(-T)) \Big)^2 \tag{19}$$

To analyze Equation (19), we present a proposition to simplify notations.

**Proposition A.1.** *The solution for real-valued variables $a, b$ in the optimization problem:*

$$\min_{a,b} \sum_{i=1}^{k} (au_i + b + c_i)^2 \tag{20}$$

*in which $u_i$'s and $c_i$'s are known constants with the constraint that $u_i$'s are not all the same, satisfies that:*

$$a \leq \zeta(\boldsymbol{u}) \max_i |c_i|, \quad b = -\frac{1}{k} \sum_{i=1}^{k} c_i - \frac{a}{k} \cdot \sum_{i=1}^{k} u_i$$

*where $\zeta(\boldsymbol{u}) = 2\sum_{i<j} |u_i - u_j| / \sum_{i<j} (u_i - u_j)^2$*

*Proof of Proposition A.1.* Let $L(a,b) = \sum_i (au_i + b + c_i)^2$ and enforce $\frac{\partial L}{\partial a} = \frac{\partial L}{\partial b} = 0$, we have:

$$a \sum_i u_i^2 + b \sum_i u_i + \sum_i c_i u_i = 0 \tag{21}$$

and

$$a \sum_i u_i + bk + \sum_i c_i = 0. \tag{22}$$

This gives:

$$a = \Big( \sum_i u_i^2 - k^{-1} \big( \sum_i u_i \big)^2 \Big)^{-1} \cdot$$

$$\Big( k^{-1} \sum_i u_i \sum_i c_i - \sum_i c_i u_i \Big)$$

$$= \Big( \sum_{i<j} (u_i - u_j)^2 \Big)^{-1} \Big( -\sum_{i<j} (u_i - u_j)(c_i - c_j) \Big)$$

$$\leq \frac{\sum_{i<j} |u_i - u_j|(|c_i| + |c_j|)}{\sum_{i<j} (u_i - u_j)^2}$$

$$\leq \frac{\sum_{i<j} |u_i - u_j|}{\sum_{i<j} (u_i - u_j)^2} \cdot 2 \max_i |c_i|$$

Therefore, $a \leq \zeta(\boldsymbol{u}) \max_i |c_i|$ and the remaining part regarding $b$ can be obtained directly from Eq. 22.

The following corollary applies Proposition A.1 to the learning of the piecewise linear coefficients.

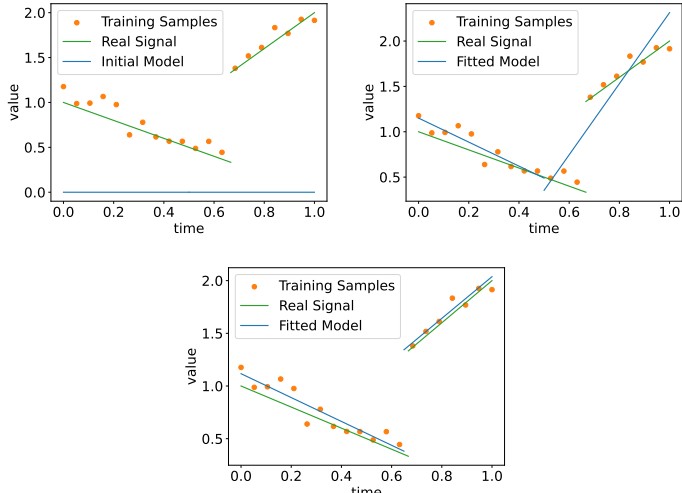

Figure 6: **(Left)** Initial model. **(Middle)** Without integral normalization, the model cannot fit piecewise linear signals with unequal segment lengths. **(Right)** After integral normalization, the model successfully fits the piecewise linear signal with unequal segment lengths.

**Corollary A.2.** *The solution $\Theta_\infty$ of the optimization problem:*

$$\underset{\Theta}{\text{minimize}} \ \mathcal{L}^T(\Theta) = \sum_{j=1}^m \left( \tilde{V}_\Theta^T(\tau_j) - v_j \right)^2 \tag{23}$$

*satisfies:*

$$\max_j \left| \tilde{V}_{\tilde{\Theta}_\infty}(\tau_j) - \tilde{V}_{\Theta^\star}(\tau_j) \right| \le O\left( \max_{1 \le j \le m} |\psi_j| \right) \tag{24}$$

*Proof of Corollary A.2.* Comparing Equation (19) and Equation (20), by setting $u_i = \tau_j$ and $c_j = -\psi_j + o(\exp(-T))$ we have:

$$m_i = m_i^\star + O\left( \max_j(|\psi_j|) + o(\exp(-T)) \right) \tag{25}$$

$$b_i = b_i^\star + O\left( \max_j(|\psi_j|) + o(\exp(-T)) \right) \tag{26}$$

given that the sampling points $\tau_j$'s are fixed (*i.e.*, $\zeta(\boldsymbol{u})$ is a constant). When $T \to \infty$, we obtain $m_i = m_i^\star + O\left( \max_j(|\psi_j|) \right)$ and $b_i = b_i^\star + O\left( \max_j(|\psi_j|) \right)$, which implies:

$$\left| \tilde{V}_{\Theta_\infty}(\tau) - \tilde{V}_{\Theta^\star(\tau)} \right| = \left| \left( m_i(\Theta_\infty) - m_i^\star \right)\tau + \left( b_i(\Theta_\infty) - b_i^\star \right) \right|$$
$$\le O\left( \max_j(|\psi_j|) \right)$$

since $0 \le \tau \le 1$.

The rest of the proof shows that under our assumptions, the sample points will eventually be classified into correct intervals.

In the following, we use $\{m_i(T)\}, \{b_i(T)\}$, and $\{t_i(T)\}$ to denote the optimal solution for the objective $\mathcal{L}^T(\Theta)$. Due to noisy samples and smoothing process, the optimal solution is different from $\Theta^\star$.

Consider $(\tau_j, v_j)$ with $t_{i-1}^* \le \tau_j < t_i^*$. Since $|t_i(\Theta_0) - t_i^\star| \le h \le |t_i^\star - t_{i-1}^\star|$, $\tau_j$ must fall into one of the intervals $[t_{i-2}(\Theta_0), t_{i-1}(\Theta_0)], [t_{i-1}(\Theta_0), t_i(\Theta_0)]$, and $[t_i(\Theta_0), t_{i+1}(\Theta_0)]$.

In the following, we discuss the stationary point at a specific temperature $T$. In practice, we can stay at the temperature $T$ until the gradient vanishes and then we move on to a higher temperature.

$$\frac{\partial w_l^{(i)}}{\partial t_{i-1}} = T \cdot w_l^{(i)^2} \cdot \exp\left(T(\tau - t_{i-1})\right)$$

$$\frac{\partial w_r^{(i)}}{\partial t_i} = -T \cdot w_r^{(i)^2} \cdot \exp\left(T(t_i - \tau)\right),$$

Setting $\frac{\partial \mathcal{L}^T}{\partial t_i} = 0$ we have:

$$
\begin{aligned}
0 = \quad & T \cdot \sum_{t_i \leq \tau_j < t_{i+1}} \left(\Delta_j w_l^{(i+1)^2} \cdot \exp(T(\tau_j - t_i))\right) \\
& \cdot \left((m_i \tau_j + b_i) - (m_{i+1} \tau_j + b_{i+1})\right) \\
& - T \cdot \sum_{t_{i-1} \leq \tau_j < t_i} \left(\Delta_j w_r^{(i)^2} \cdot \exp(T(t_i - \tau_j))\right) \\
& \cdot \left((m_{i+1} \tau_j + b_{i+1}) - (m_i \tau_j + b_i)\right)
\end{aligned}
$$

where $\Delta_j$ is defined by:

$$\Delta_j := \tilde{V}_\Theta^T(\tau_j) - v_j$$

Thus, we have:

$$
\begin{aligned}
& \sum_{t_i \leq \tau_j < t_{i+1}} \left(\Delta_j w_l^{(i+1)^2} \cdot \exp(T(\tau_j - t_i))\right) \\
& \cdot \left((m_i \tau_j + b_i) - (m_{i+1} \tau_j + b_{i+1})\right) \\
= & \sum_{t_{i-1} \leq \tau_j < t_i} \left(\Delta_j w_r^{(i)^2} \cdot \exp(T(t_i - \tau_j))\right) \\
& \cdot \left((m_{i+1} \tau_j + b_{i+1}) - (m_i \tau_j + b_i)\right),
\end{aligned}
$$

or equivalently,

$$
\begin{aligned}
& \exp(-Tt_i) \sum_{t_i \leq \tau_j < t_{i+1}} \left(\Delta_j w_l^{(i+1)^2} \cdot \exp(T\tau_j)\right) \\
& \cdot \left((m_i \tau_j + b_i) - (m_{i+1} \tau_j + b_{i+1})\right) \\
= & \exp(Tt_i) \sum_{t_{i-1} \leq \tau_j < t_i} \left(\Delta_j w_r^{(i)^2} \cdot \exp(-T\tau_j)\right) \\
& \cdot \left((m_{i+1} \tau_j + b_{i+1}) - (m_i \tau_j + b_i)\right). \qquad (27)
\end{aligned}
$$

Consider the case where $t_i$ is not currently at the correct location. Without loss of generality, we assume $t_i$ is too far to the right. This implies that there exists $\tau_j$ satisfying $t_{i-1} \leq t_i^\star \leq \tau_j < t_i$. Suppose:

$$t_{i-1} \leq \tau_{j_0} < \tau_{j_0+1} < \cdots < \tau_{j-1} < t_i^\star$$

and

$$t_i^\star \leq \tau_j < t_i \leq \tau_{j+1} < \cdots < \tau_{j_1} < t_{i+1}.$$

Since $\tau_{j+1}, \ldots, \tau_{j_1}$ all fall inside the correct interval, we can fully recover the corresponding segment and have $\Delta_q = O(e^{-T})$ for $q = j + 1, \ldots, j_1$. The left-hand side of Equation (27) is, therefore, $O(\exp(-T))$, which enforces the right-hand side of the equation to keep getting smaller. Since $(\tau_j, v_j)$ is not consistent with $\tau_{j_0}, \ldots \tau_{j-1}$ and $\Delta$'s are the errors of linear regression, we claim that $\Delta_r, r = j_0, \ldots, j - 1$ would never be close to zero. To make Equation (27) hold true, $t_i$ has to be decreased. This process will last until $t_i < \tau_j$ (i.e., the sample point $(\tau_j, v_j)$ is excluded from the wrong interval $[t_{i-1}, t_i]$).

In this way, we prove that eventually all sample points will be classified into correct intervals, thereby reducing to the base case that has been proved earlier.

$$\square$$

Table 6: Steering prediction accuracy when PPLNs have different numbers of line segments.

| Segment Count | 3 | 6 |
|---|---|---|
| RMSE $\downarrow$ | $3.15 \pm 0.081$ | $3.23 \pm 0.200$ |
| EVA $\uparrow$ | $0.926 \pm 0.004$ | $0.923 \pm 0.010$ |
| Segment Count | 9 | 12 |
| RMSE $\downarrow$ | $3.44 \pm 0.141$ | $3.34 \pm 0.148$ |
| EVA $\uparrow$ | $0.913 \pm 0.007$ | $0.917 \pm 0.008$ |

Table 7: We use ablation studies to demonstrate the practical implication of the smoothing operator.

| Smoothing? | Motion Deblurring | | | Steering Prediction | | Human Pose Estimation | | |
|---|---|---|---|---|---|---|---|---|
| | MSE $\downarrow$ | PSNR $\uparrow$ | SSIM $\uparrow$ | RMSE $\downarrow$ | EVA $\uparrow$ | 2D-2 $\downarrow$ | 2D-3 $\downarrow$ | 3D $\downarrow$ |
| ✗ | 0.157 | 23.994 | 0.718 | $\mathbf{3.07 \pm 0.100}$ | $\mathbf{0.930 \pm 0.004}$ | **6.58** | **6.45** | **71.64** |
| ✓ | **0.152** | **24.284** | **0.728** | $3.15 \pm 0.081$ | $0.926 \pm 0.004$ | 6.76 | 6.51 | 73.05 |

## A.10 Justifying integral normalization

Recall that when we construct a PPLN node without integral normalization or smoothing, the output $\tilde{V}_\Theta(t)$ is given as:

$$\tilde{V}_\Theta(t) = \begin{cases} m_1 t + b_1 & t_0 \leq t < t_1 \\ m_2 t + b_2 & t_1 \leq t < t_2 \\ \dots \\ m_n t + b_n & t_{n-1} \leq t \leq t_n \end{cases} \tag{28}$$

With $\mathbf{m} = (m_1, \dots, m_n)^T$, $\mathbf{b} = (b_1, \dots, b_n)^T$, and $\mathbf{t} = (t_0, \dots, t_n)^T$, we have:

$$\frac{\partial \tilde{V}_\Theta(t_k)}{\partial \mathbf{m}} = (0, \cdots, 0, t_k, 0, \cdots, 0) \tag{29}$$

$$\frac{\partial \tilde{V}_\Theta(t_k)}{\partial \mathbf{b}} = (0, \cdots, 0, 1, 0, \cdots, 0) \tag{30}$$

$$\frac{\partial \tilde{V}_\Theta(t_k)}{\partial \mathbf{t}} = \mathbf{0} \tag{31}$$

at the input timestamp $t_0 \leq t_k \leq t_n$.

We make two observations here. First, both $\frac{\partial \tilde{V}_\Theta(t_k)}{\partial \mathbf{m}}$ and $\frac{\partial \tilde{V}_\Theta(t_k)}{\partial \mathbf{b}}$ are very sparse vectors with only one non-zero entry corresponding to the specific segment $t_k$ belongs to. At training time, it is natural to assume the training timestamps are uniformly distributed across $[t_0, t_n]$. This means the amount of training data is proportional to the segment length. Long segments receive intensive training because there are a lot of available training examples. Short segments receive little training because there are only a limited number of training examples falling inside their ranges. The imbalance presents an instability risk because different segments are learning at different rates. Additionally, $\frac{\partial \tilde{V}_\Theta(t_k)}{\partial \mathbf{t}}$ is an all-zero vector. A gradient-based optimizer is therefore unable to adjust the segment lengths. This means the segment endpoints are hard-coded positions instead of trainable parameters.

To illustrate the second observation, consider the toy example of a 2-segment linear signal, as shown in Figure 6. The green real signal generates a set of orange training samples with measurement noises. Assuming no prior knowledge of the real signal besides that it has two segments, we initialize a 2-segment parametric model with zero slopes, zero intercepts, and equal segment lengths (Figure 6 (Left)). After training, the node converges to the model shown in Figure 6 (Middle). The second segment deviates from the real signal significantly because it attempts to accommodate the last several training samples in the first segment.

As discussed in the body, we use the integral normalization operator $\sigma(\cdot)$ to address the above challenges:

$$\sigma(\tilde{V}_\Theta(t)) = \tilde{V}_\Theta(t) - \int_0^1 \tilde{V}_\Theta(t) dt + \overline{V} \tag{32}$$

where $\overline{V}$ is a parameter that controls the mean of $\tilde{V}_\Theta(t)$ when $0 \leq t \leq 1$.

This gives:

$$\frac{\partial \sigma(\tilde{V}_\Theta(t_k))}{\partial \mathbf{m}} = \frac{\partial \tilde{V}_\Theta(t_k)}{\partial \mathbf{m}}$$
$$- \frac{1}{2}\left(t_1^2 - t_0^2, \cdots, t_k^2 - t_{k-1}^2, \cdots, t_n^2 - t_{n-1}^2\right) \tag{33}$$

$$\frac{\partial \sigma(\tilde{V}_\Theta(t_k))}{\partial \mathbf{b}} = \frac{\partial \tilde{V}_\Theta(t_k)}{\partial \mathbf{b}}$$
$$- (t_1 - t_0, \cdots, t_k - t_{k-1}, \cdots, t_n - t_{n-1}) \tag{34}$$

$$\frac{\partial \sigma(\tilde{V}_\Theta(t_k))}{\partial \mathbf{t}} = (\cdots, m_{k+1}t_k + b_{k+1} - m_k t_k - b_k, \cdots) \tag{35}$$

where all three gradient vectors contain rich gradient information in every element, encouraging a smooth and swift convergence. Figure 6 (Right) illustrates how the model accurately approximates both linear segments in the toy example after incorporating integral normalization.

## A.11 Ablation studies: number of line segments

As shown in Table 6, we fail to observe noticeable performance improvement when increasing the number of line segments $n$ in the formulation. Therefore, we decide to use $n = 3$ in all the experiments discussed in the body of this paper for better efficiency.

## A.12 Ablation studies: smoothing

As shown in Table 7, the smoothing operator has an insignificant impact on the prediction quality. However, we point out several facts related to the smoothing operator as a guideline for potential future applications. First, the smoothing operator allows us to have a simple model (Theorem 1) with analytical analyzable properties. Second, the smoothing operator does not introduce any additional trainable parameters. Third, when the size of the network is relatively small (*i.e.*, steering prediction and human pose estimation), our results suggest that using PPLNs without smoothing is slightly better. Finally, when the size of the network is large (*i.e.*, motion deblurring), smoothing introduces a small performance improvement.

## A.13 Impact Statement

This paper presents work whose goal is to advance the field of Machine Learning. There are many potential societal consequences of our work, none of which we feel must be specifically highlighted here. Regarding the environmental impact, our experiments are conducted on an internal GPU cluster, and the total emission is estimated to be 81.12 $kgCO_2$eq, equivalent to 328 km driven by an average car. This emission estimation is conducted using the Machine Learning Impact calculator presented by Lacoste et al.. To mitigate repetitive labor and negative environmental impact in future research, we have released our open-source implementation together with trained network weights after the anonymous period.

