# OpenReview forum: "PPLNs: Parametric Piecewise Linear Networks for Event-Based Temporal Modeling and Beyond"
_NeurIPS.cc/2024/Conference — NeurIPS 2024 poster_

### Official Review · Reviewer_oEKu · 2024-07-10

**Soundness:** 3
**Presentation:** 3
**Contribution:** 3
**Rating:** 5
**Confidence:** 4

**Summary:**

This paper proposes Parametric Piecewise Linear Networks (PPLNs) for event-based temporal modeling, which emulate biological principles by representing membrane potentials as parametric mappings. The authors demonstrate how a straightforward modification enables standard multi-layer perceptrons and convolution operators to accommodate this parameterization of membrane potential. Experimental results showcase that the proposed PPLNs attain state-of-the-art performance in typical event-based vision tasks, including steering prediction, human pose estimation, and motion deblurring.

**Strengths:**

i) The topic of bio-inspired parametric piecewise linear networks for temporal modeling is very novel and attractive.

ii) The authors sufficient experiments in the main paper and the supplemental material to help reader better understand the main contributions of this work.

iii) The writing is straightforward, clear, and easy to understand.

**Weaknesses:**

i) Could you give the time complexity of PPLNs, or the inference time on the three downstream tasks, such as steering prediction, human pose estimation, and motion deblurring?

ii) Which hyperparameters have the greatest influence on LLPNs? Please explain the analysis from an experimental perspective.

**Questions:**

Please see the weakness and response each comment.

**Limitations:**

There is a lack of analysis of the computational complexity of PPLNs, including experimental tests on the inference time of three downstream event-based vision tasks.

---

> ### Author Rebuttal · Authors · 2024-08-04
>
> Dear Reviewer,
>
> We appreciate the discussion with you previously in our submission to another venue. We have answered the questions you listed earlier but did not receive your response. May we ask whether you think our rebuttal has clarified your concerns? If so, we kindly encourage you to consider raising the rating. Thanks!
>
> **Comment:** Could you give the time complexity of PPLNs, or the inference time on the three downstream tasks, such as steering prediction, human pose estimation, and motion deblurring?
>
> **Response:** Thank you for your comment! We appreciate the opportunity to provide additional information. Below is a table comparing various performance indicators between our PPLN approach and baseline methods for different tasks. All timing measurements were conducted on the same Titan V server in our institution. For motion deblurring, the baseline approach is U-Net, which has an identical network architecture to PPLNs constructed from regular convolutional layers. For steering prediction and human pose estimation, the baseline approaches are "Calabrese" and "Hu", respectively.
>
> Importantly, we emphasize that in steering prediction and human pose estimation, PPLNs output ten times as much information as the baseline, which contributes to their slower performance in these two tasks.
>
> | Task | PPLN | Baseline |
> |---------------------|-----------------------------------|-----------------------------------|
> | Motion Deblurring | Training: 30 minutes for 50 epochs | Training: 24 minutes for 50 epochs |
> | | Inference: 16.75 blurry frames/s | Inference: 19.16 blurry frames/s |
> | | Parameters: 192,095,064 | Parameters: 172,622,332 |
> | Steering Prediction | Training: 16.8 hours for 200 epochs | Training: 8.2 hours for 200 epochs |
> | | Inference: 0.27s/iteration | Inference: 0.11s/iteration |
> | | Parameters: 455,338 | Parameters: 463,425 |
> | Human Pose Estimation| Training: 3.1 hours for 20 epochs | Training: 1.3 hours for 20 epochs |
> | | Inference: 0.32s/iteration | Inference: 0.18s/iteration |
> | | Parameters: 215,648 | Parameters: 218,592 |
>
> **Comment:** Which hyperparameters have the greatest influence on LLPNs? Please explain the analysis from an experimental perspective.
>
> **Response:** Thank you for your comment! We appreciate the opportunity to address this question from an experimental perspective. Our analysis indicates that the integral normalization constant has the greatest influence on PPLNs, as demonstrated in Table 5. Particularly in motion deblurring tasks, where the ground-truth constant value is available, the MSE improvement attributable to integral normalization can be as high as 42.4%. In tasks such as steering prediction and human pose estimation, where the constant value must be predicted, integral normalization still enhances performance but to a lesser extent. This result demonstrates the importance of using a good value as the integral normalization constant.
>
> Furthermore, our experiments reveal that the number of line segments per layer also impacts prediction quality, as evidenced by Table 7 in the paper. We recommend utilizing three segments per layer as it strikes a balance between quality and parameter count, resulting in optimal performance.
>
> Sincerely,
>
> Authors

---

> > ### Comment · Reviewer_oEKu · 2024-08-13
> >
> > The author has addressed my concerns, and I will maintain the original score. I hope the author will incorporate the  different  experimental tasks into the appendix of the camera-ready version.

---

> > > ### Author Response · Authors · 2024-08-13
> > >
> > > Thanks for letting us know, and we are glad to hear the rebuttal has addressed your concerns! We will make sure to include the appendix as part of the camera-ready version. To enhance the chance of acceptance, may we ask if it is possible for you to increase the rating to weak accept (or above)?

---

### Official Review · Reviewer_4tcq · 2024-07-13

**Soundness:** 4
**Presentation:** 3
**Contribution:** 4
**Rating:** 7
**Confidence:** 5

**Summary:**

1. The paper presents Parametric Piecewise Linear Networks (PPLNs), a novel neural network architecture inspired by neuromorphic principles for temporal vision inference.

2. The innovative approach of PPLNs lies in modeling the membrane potential of artificial neurons as parametric piecewise linear functions with learnable coefficients, echoing the design of Kolmogorov-Arnold Networks (KANs) but with input-dependent adaptability.

3. Experimental results demonstrate PPLNs' state-of-the-art performance in event-based vision applications such as steering prediction, human pose estimation, and motion deblurring, outperforming existing methods with improved accuracy and efficiency.

**Strengths:**

1. PPLNs are inspired by neuromorphic principles, mimicking the behavior of biological neurons, which adds a layer of biological plausibility to their computational model.

2. The authors provide a thorough evaluation, including ablation studies, which helps in understanding the contribution of different components of PPLNs to the overall performance. The experiments showcase PPLNs achieving state-of-the-art results across various vision tasks, highlighting the effectiveness of the model in processing event-based data. The paper also evaluates PPLNs on conventional frame-based tasks, showing the model's generalizability beyond just event-based applications.

**Weaknesses:**

1. The authors did not compare with some of the latest event-based deblurring algorithms, such as REFID[1].

2. In addition, the authors should provide a direct comparative experiment using a KAN-based backbone to further demonstrate the superiority of the proposed solution.

[1] Sun, Lei, et al. "Event-based frame interpolation with ad-hoc deblurring." Proceedings of the IEEE/CVF Conference on Computer Vision and Pattern Recognition. 2023.

**Questions:**

1. The piecewise linear nature of this paper is akin to an input-dependent activation function. An SNN using multi-layer LIF neurons with learnable parameters also seems to achieve a similar level of biological plausibility. Could the authors provide further explanation from this perspective?

2. Based on Figure 2d, I am still unclear whether the input t for PPLN is an external query (you can choose your desire timestamp t) or an inherent timestamp of the event. If it is the former, does this mean that all neurons query with the same input t? If it is the latter, how is this implemented? Given that there are many input events with numerous timestamps, how does the method handle this?

A good response to the Weaknesses and Questions will improve my initial rating.

**Limitations:**

The authors discussed the environmental impact of the training phase for their proposed solution.

---

> ### Author Rebuttal · Authors · 2024-08-04
>
> Dear Reviewer,
>
> Here are the responses to the concerns in your review. We look forward to your additional input during the reviewer-author discussion period. While we are grateful for the weak accept vote, we encourage you to consider raising the rating if you deem it appropriate.
>
> **Comment:** The authors did not compare with some of the latest event-based deblurring algorithms, such as REFID[1].
>
> **Response:** Thank you for your comment! We recognize that REFID is an important work in event vision. However, REFID takes two blurry frames as input and performs frame interpolation. The design of REFID relies on the fact that there are two conventional frames available as input, which deviates from our setting that assumes only one input frame. We will make sure to discuss REFID and why it is not included as a baseline approach in our revision.
>
> **Comment:** In addition, the authors should provide a direct comparative experiment using a KAN-based backbone to further demonstrate the superiority of the proposed solution.
>
> **Response:** Thank you for your comment! Unfortunately, there are two reasons why we are unable to provide such a direct comparative experiment. First, KANs do not have temporal modeling, and there is not yet consensus on how KANs can be modified to support temporal modeling. Additionally, KANs are very slow (c.f., page 33 of 2404.19756). This is why KANs are experimented on very simple toy datasets. Sadly, we do not have access to the computational resources to evaluate KANs on larger-scale applications discussed in our submission.
>
> **Comment:** The piecewise linear nature of this paper is akin to an input-dependent activation function. An SNN using multi-layer LIF neurons with learnable parameters also seems to achieve a similar level of biological plausibility. Could the authors provide further explanation from this perspective?
>
> **Response:** Thank you for your comment! We agree that the proposed PPLN is conceptually very similar to the SNN. However, we highlight four unique characteristics of PPLNs. First, PPLN focuses on representing the membrane potentials instead of the interconnection between artificial neurons. Second, PPLN models the membrane potential using a real value instead of explicit binary spikes. Third, PPLN does not restrict the sign of the slope. Lastly, PPLN does not explicitly enforce any line segment to be flat. For more details, we encourage you to check out the discussion in Section A.4 of the supplementary material.
>
> **Comment:** Based on Figure 2d, I am still unclear whether the input t for PPLN is an external query (you can choose your desire timestamp t) or an inherent timestamp of the event. If it is the former, does this mean that all neurons query with the same input t? If it is the latter, how is this implemented? Given that there are many input events with numerous timestamps, how does the method handle this?
>
> **Response:** Thank you for your comment! We appreciate the opportunity to clarify. The model input consists of two parts, the non-temporal component $\mathbf{x}$, and the temporal component $t$. For each $\mathbf{x}$, we typically hope to make inferences at multiple timestamp $t$'s. For example, we are interested in the sharp frames at different timestamps $t \in [0, 1]$ that correspond to the input blurry image and events ($\mathbf{x}$). From this perspective, the input $t$ is an external query, and all the neurons receive the same $t$ value.
>
> **Comment:** A good response to the Weaknesses and Questions will improve my initial rating.
>
> **Response:** We sincerely appreciate your willingness to improve the rating! Please let us know if our response has addressed your concerns. We look forward to discussing the submission more with you.
>
> Sincerely,
>
> Authors

---

> > ### Comment · Reviewer_4tcq · 2024-08-13
> >
> > Thank you for the enthusiastic reply. I understand that PPLN is a network with each neuron with a dynamic equation, which can be queried through t, and all neurons receive the same input t. This may sound a bit weird because the execution of neurons in different layers of a neural network is sequential rather than parallel. But I think it is still biologically plausible, as the dynamic equations of each layer are different, which in turn means that the 't' for each layer is different. Does the author have any further insights?
> >
> > Overall, after reading the opinions of other reviewers, I still think this paper presents some interesting insights. I am willing to raise my rating to increase the likelihood of this paper receiving more attention within the community.

---

> > > ### Author Response · Authors · 2024-08-13
> > >
> > > Thanks for rasing the score and the follow-up discussion! We agree that using different $t$'s in different layers is a biologically plausible design. In fact, $t$ does not even have to be part of the input. The previous layer can predict a timestamp for the next layer. We will make sure to discuss this in our revision.

---

### Official Review · Reviewer_NNdg · 2024-07-13

**Soundness:** 2
**Presentation:** 3
**Contribution:** 3
**Rating:** 5
**Confidence:** 4

**Summary:**

This paper introduces Parametric Piecewise Linear Networks (PPLNs), a novel approach to temporal modeling inspired by biological neural principles. PPLNs represent neuron membrane potentials as piecewise linear functions with learnable coefficients, aiming to allow for explicit temporal modeling. The authors evaluate PPLNs on three event-based vision tasks: motion deblurring, steering prediction, and human pose estimation, claiming significant improvements over baseline models and some state-of-the-art methods. The paper also attempts to show that PPLNs can generalize to conventional frame-based versions of these tasks.

**Strengths:**

1.	The PPLN approach presents an interesting attempt to model temporal dynamics in neural networks, drawing inspiration from biological neural systems.
2.	The paper evaluates the proposed method on multiple event-based vision tasks, providing a reasonably broad assessment of its performance.

**Weaknesses:**

1.	The paper lacks a thorough analysis of the computational costs associated with PPLNs. Without this information, it's difficult to assess the practical viability of the approach, especially in comparison to existing methods.
2.	While the authors claim biological inspiration, the paper does not adequately explore how closely PPLNs actually mimic neuronal behavior. The connection to biological systems seems superficial and not well-substantiated.
3.	The paper's claims about PPLNs' applicability to conventional frame-based tasks are not adequately supported. The limited experiments in this domain do not provide convincing evidence of the method's broader applicability beyond event-based vision.
4.	The paper fails to provide a detailed analysis of how sensitive PPLNs are to hyperparameter choices, such as the number of line segments in the parameterization. This omission raises questions about the robustness and reproducibility of the results.

**Questions:**

1.	What is the exact computational overhead of PPLNs compared to traditional neural network layers?
2.	Can you provide a detailed sensitivity analysis for the key hyperparameters of PPLNs?
3.	Can you provide more evidence to support the biological plausibility of PPLNs beyond the initial inspiration?

**Limitations:**

The authors have addressed some limitations of their work, particularly in the supplementary material. However, a more comprehensive discussion of potential negative societal impacts and failure cases would be beneficial. The paper could also benefit from a more detailed analysis of the computational requirements and scalability of PPLNs.

---

> ### Author Rebuttal · Authors · 2024-08-04
>
> Dear Reviewer,
>
> Here are the responses to the concerns in your review. We look forward to your additional input during the discussion period. Meanwhile, we encourage you to consider raising the rating if you deem it appropriate.
>
> **Comment:** The paper lacks a thorough analysis of the computational costs associated with PPLNs. Without this information, it's difficult to assess the practical viability of the approach, especially in comparison to existing methods.
>
> **Question:** What is the exact computational overhead of PPLNs compared to traditional neural network layers?
>
> **Response:** Thank you for your comment and question! We analyze the computational cost of PPLNs and the baseline approaches in Section A.8 of the supplementary material. For your convenience, below is a table comparing various performance indicators between our PPLN approach and baseline methods for different tasks. All timing measurements were conducted on the same Titan V server in our institution. For motion deblurring, the baseline approach is U-Net, which has an identical network architecture to PPLNs constructed from regular convolutional layers. For steering prediction and human pose estimation, the baseline approaches are "Calabrese" and "Hu", respectively.
>
> Importantly, we emphasize that in steering prediction and human pose estimation, PPLNs output ten times as much information as the baseline, which contributes to their slower performance in these two tasks.
>
> | Task | PPLN | Baseline |
> |---------------------|-----------------------------------|-----------------------------------|
> | Motion Deblurring | Training: 30 minutes for 50 epochs | Training: 24 minutes for 50 epochs |
> | | Inference: 16.75 blurry frames/s | Inference: 19.16 blurry frames/s |
> | | Parameters: 192,095,064 | Parameters: 172,622,332 |
> | Steering Prediction | Training: 16.8 hours for 200 epochs | Training: 8.2 hours for 200 epochs |
> | | Inference: 0.27s/iteration | Inference: 0.11s/iteration |
> | | Parameters: 455,338 | Parameters: 463,425 |
> | Human Pose Estimation| Training: 3.1 hours for 20 epochs | Training: 1.3 hours for 20 epochs |
> | | Inference: 0.32s/iteration | Inference: 0.18s/iteration |
> | | Parameters: 215,648 | Parameters: 218,592 |
>
> **Comment:** While the authors claim biological inspiration, the paper does not adequately explore how closely PPLNs actually mimic neuronal behavior. The connection to biological systems seems superficial and not well-substantiated.
>
> **Question:** Can you provide more evidence to support the biological plausibility of PPLNs beyond the initial inspiration?
>
> **Response:** Thank you for your comment and question! In Section A.4 of the supplementary, we discuss how PPLNs are connected with the biological neuromorphic mechanism in detail. In particular, we highlight that PPLNs focus on an explicit parameterization of the membrane potential, which is underexplored in existing work. While they do not follow all neural principles in biology, we hope you agree that PPLNs are closer to real neural systems than conventional artificial neural networks.
>
> **Comment:** The paper's claims about PPLNs' applicability to conventional frame-based tasks are not adequately supported. The limited experiments in this domain do not provide convincing evidence of the method's broader applicability beyond event-based vision.
>
> **Response:** Thank you for your comment! We agree with the reviewer that the experiments in the paper do not justify the use of PPLNs in conventional frame-based tasks. In fact, for most conventional frame-based tasks, we would advocate against the use of PPLNs. As discussed in the paragraph starting at line 195, most conventional frame-based tasks, such as object detection and segmentation, are associated with sufficient high-quality training data. PPLNs are most applicable to scenarios when the training data is limited, causing modeling to play a more important role. Such scenarios include most tasks in event-based vision, as well as very niche situations in conventional frame-based vision.
>
> **Comment:** The paper fails to provide a detailed analysis of how sensitive PPLNs are to hyperparameter choices, such as the number of line segments in the parameterization. This omission raises questions about the robustness and reproducibility of the results.
>
> **Question:** Can you provide a detailed sensitivity analysis for the key hyperparameters of PPLNs?
>
> **Response:** Thank you for your comment and question! We appreciate the opportunity to clarify. Our analysis indicates that the integral normalization constant has the greatest influence on PPLNs, as demonstrated in Table 3. Particularly in motion deblurring tasks, where the ground-truth constant value is available, the MSE improvement attributable to integral normalization can be as high as 42.4%. In tasks such as steering prediction and human pose estimation, where the constant value must be predicted, integral normalization still enhances performance but to a lesser extent. This result demonstrates the importance of using a good value as the integral normalization constant.
>
> Furthermore, our experiments reveal that the number of line segments per layer also impacts prediction quality, as evidenced by Table 7 in the supplementary material. We recommend utilizing three segments per layer as it strikes a balance between quality and parameter count, resulting in optimal performance.
>
> **Comment**: However, a more comprehensive discussion of potential negative societal impacts and failure cases would be beneficial.
>
> **Response**: Thanks for your comment! We fail to observe a noticeable improvement when having sufficient high-quality training data, which is rarely the case in event-based applications. However, this impedes PPLNs from being applied widely to conventional vision tasks. A promising direction is to integrate PPLNs into Parameter-Efficient Fine-Tuning (PEFT) techniques to fine-tune model parameters on small datasets.
>
> Sincerely,
>
> Authors

---

> > ### Comment · Reviewer_NNdg · 2024-08-13
> >
> > I appreciate the response, which has addressed my initial concerns. I will accordingly adjust my rating.

---

> > > ### Author Response · Authors · 2024-08-13
> > >
> > > Thank you very much for your reply and increasing the rating!

---

### Author Rebuttal · Authors · 2024-08-04

We would like to express our sincere gratitude to the reviewers for their thoughtful evaluation of our submission. While we appreciate recognition from all reviewers, we believe there may be additional aspects of our research that warrant further consideration. Specifically, we feel that Theorem 3.1, which theoretically analyzes the convergence properties of PPLNs, has not received sufficient attention. We also encourage the reviewers to check out our supplementary material, which discusses the biological principles and the hyperparameter choices. In the responses below, we aim to address any remaining concerns and highlight the strengths of our contribution. We welcome further discussion and clarification and are committed to addressing any lingering questions to the best of our ability.

---

### Author Response · Authors · 2024-08-12
**Reviewer-Author Discussion Period**

Dear Reviewers,

Thank you for your constructive feedback and valuable suggestions! As a gentle reminder, there are just under three days remaining before the reviewer-author discussion period ends. We would like to seize this opportunity to address any questions or concerns you may still have regarding our submission. Furthermore, we encourage you to consider raising the rating if you deem it appropriate.

Many thanks in advance!

Sincerely,

Authors

---

### Decision · Program_Chairs · 2024-09-25

**Decision:**

Accept (poster)

**Comment:**

The reviewers recognize the proposed approach as a novel and interesting approach to event-based vision processing, with a convincing demonstration of its performance, including a number of ablation studies. There were some discussions about inference time, hyperparameters and biological plausibility, which the authors could address for the most part during the discussion period. Overall, there is concensus that the paper makes a solid contribution and should be accepted.